# Fast Convergence of Regularized Learning in Games

**Vasilis Syrgkanis**
Microsoft Research
New York, NY
vasy@microsoft.com

**Alekh Agarwal**
Microsoft Research
New York, NY
alekha@microsoft.com

**Haipeng Luo**
Princeton University
Princeton, NJ
haipengl@cs.princeton.edu

**Robert E. Schapire**
Microsoft Research
New York, NY
schapire@microsoft.com

## Abstract

We show that natural classes of regularized learning algorithms with a form of recency bias achieve faster convergence rates to approximate efficiency and to coarse correlated equilibria in multiplayer normal form games. When each player in a game uses an algorithm from our class, their individual regret decays at $O(T^{-3/4})$, while the sum of utilities converges to an approximate optimum at $O(T^{-1})$–an improvement upon the worst case $O(T^{-1/2})$ rates. We show a black-box reduction for any algorithm in the class to achieve $\tilde{O}(T^{-1/2})$ rates against an adversary, while maintaining the faster rates against algorithms in the class. Our results extend those of Rakhlin and Shridharan [17] and Daskalakis et al. [4], who only analyzed two-player zero-sum games for specific algorithms.

## 1 Introduction

What happens when players in a game interact with one another, all of them acting independently and selfishly to maximize their own utilities? If they are smart, we intuitively expect their utilities — both individually and as a group — to grow, perhaps even to approach the best possible. We also expect the dynamics of their behavior to eventually reach some kind of equilibrium. Understanding these dynamics is central to game theory as well as its various application areas, including economics, network routing, auction design, and evolutionary biology.

It is natural in this setting for the players to each make use of a no-regret learning algorithm for making their decisions, an approach known as *decentralized no-regret dynamics*. No-regret algorithms are a strong match for playing games because their regret bounds hold even in adversarial environments. As a benefit, these bounds ensure that each player's utility approaches optimality. When played against one another, it can also be shown that the sum of utilities approaches an approximate optimum [2, 18], and the player strategies converge to an equilibrium under appropriate conditions [6, 1, 8], at rates governed by the regret bounds. Well-known families of no-regret algorithms include multiplicative-weights [13, 7], Mirror Descent [14], and Follow the Regularized/Perturbed Leader [12]. (See [3, 19] for excellent overviews.) For all of these, the average regret vanishes at the worst-case rate of $O(1/\sqrt{T})$, which is unimprovable in fully adversarial scenarios.

However, the players in our setting are facing other similar, predictable no-regret learning algorithms, a chink that hints at the possibility of improved convergence rates for such dynamics. This was first observed and exploited by Daskalakis et al. [4]. For two-player zero-sum games, they developed a decentralized variant of Nesterov's accelerated saddle point algorithm [15] and showed that each player's average regret converges at the remarkable rate of $O(1/T)$. Although the resulting

dynamics are somewhat unnatural, in later work, Rakhlin and Sridharan [17] showed surprisingly that the same convergence rate holds for a simple variant of Mirror Descent with the seemingly minor modification that the last utility observation is counted twice.

Although major steps forward, both these works are limited to two-player zero-sum games, the very simplest case. As such, they do not cover many practically important settings, such as auctions or routing games, which are decidedly not zero-sum, and which involve many independent actors.

In this paper, we vastly generalize these techniques to the practically important but far more challenging case of arbitrary multi-player normal-form games, giving natural no-regret dynamics whose convergence rates are much faster than previously possible for this general setting.

**Contributions.** We show that the average welfare of the game, that is, the sum of player utilities, converges to approximately optimal welfare at the rate $O(1/T)$, rather than the previously known rate of $O(1/\sqrt{T})$. Concretely, we show a natural class of regularized no-regret algorithms with recency bias that achieve welfare at least $(\lambda/(1 + \mu))\text{OPT} - O(1/T)$, where $\lambda$ and $\mu$ are parameters in a smoothness condition on the game introduced by Roughgarden [18]. For the same class of algorithms, we show that each individual player's average regret converges to zero at the rate $O\left(T^{-3/4}\right)$. Thus, our results entail an algorithm for computing coarse correlated equilibria in a decentralized manner with significantly faster convergence than existing methods.

We additionally give a black-box reduction that preserves the fast rates in favorable environments, while robustly maintaining $\tilde{O}(1/\sqrt{T})$ regret against any opponent in the worst case.

Even for two-person zero-sum games, our results for general games expose a hidden generality and modularity underlying the previous results [4, 17]. First, our analysis identifies stability and recency bias as key structural ingredients of an algorithm with fast rates. This covers the Optimistic Mirror Descent of Rakhlin and Sridharan [17] as an example, but also applies to optimistic variants of Follow the Regularized Leader (FTRL), including dependence on arbitrary weighted windows in the history as opposed to just the utility from the last round. Recency bias is a behavioral pattern commonly observed in game-theoretic environments [9]; as such, our results can be viewed as a partial theoretical justification. Second, previous approaches in [4, 17] on achieving both faster convergence against similar algorithms while at the same time $\tilde{O}(1/\sqrt{T})$ regret rates against adversaries were shown via ad-hoc modifications of specific algorithms. We give a black-box modification which is not algorithm specific and works for all these optimistic algorithms.

Finally, we simulate a 4-bidder simultaneous auction game, and compare our optimistic algorithms against Hedge [7] in terms of utilities, regrets and convergence to equilibria.

## 2   Repeated Game Model and Dynamics

Consider a static game $G$ among a set $N$ of $n$ players. Each player $i$ has a strategy space $S_i$ and a utility function $u_i : S_1 \times \ldots \times S_n \to [0, 1]$ that maps a strategy profile $\mathbf{s} = (s_1, \ldots, s_n)$ to a utility $u_i(\mathbf{s})$. We assume that the strategy space of each player is finite and has cardinality $d$, i.e. $|S_i| = d$. We denote with $\mathbf{w} = (\mathbf{w}_1, \ldots, \mathbf{w}_n)$ a profile of mixed strategies, where $\mathbf{w}_i \in \Delta(S_i)$ and $w_{i,x}$ is the probability of strategy $x \in S_i$. Finally let $U_i(\mathbf{w}) = \mathbb{E}_{\mathbf{s} \sim \mathbf{w}}[u_i(\mathbf{s})]$, the expected utility of player $i$.

We consider the setting where the game $G$ is played repeatedly for $T$ time steps. At each time step $t$ each player $i$ picks a mixed strategy $\mathbf{w}_i^t \in \Delta(S_i)$. At the end of the iteration each player $i$ observes the expected utility he would have received had he played any possible strategy $x \in S_i$. More formally, let $u_{i,x}^t = \mathbb{E}_{\mathbf{s}_{-i} \sim \mathbf{w}_{-i}^t}[u_i(x, \mathbf{s}_{-i})]$, where $\mathbf{s}_{-i}$ is the set of strategies of all but the $i^{th}$ player, and let $\mathbf{u}_i^t = (u_{i,x}^t)_{x \in S_i}$. At the end of each iteration each player $i$ observes $\mathbf{u}_i^t$. Observe that the expected utility of a player at iteration $t$ is simply the inner product $\langle \mathbf{w}_i^t, \mathbf{u}_i^t \rangle$.

**No-regret dynamics.** We assume that the players each decide their strategy $\mathbf{w}_i^t$ based on a vanishing regret algorithm. Formally, for each player $i$, the regret after $T$ time steps is equal to the maximum gain he could have achieved by switching to any other fixed strategy:

$$r_i(T) = \sup_{\mathbf{w}_i^* \in \Delta(S_i)} \sum_{t=1}^{T} \left\langle \mathbf{w}_i^* - \mathbf{w}_i^t, \mathbf{u}_i^t \right\rangle.$$

The algorithm has vanishing regret if $r_i(T) = o(T)$.

**Approximate Efficiency of No-Regret Dynamics.**  We are interested in analyzing the average welfare of such vanishing regret sequences. For a given strategy profile $\mathbf{s}$ the social welfare is defined as the sum of the player utilities: $W(\mathbf{s}) = \sum_{i \in N} u_i(\mathbf{s})$. We overload notation to denote $W(\mathbf{w}) = \mathbb{E}_{\mathbf{s} \sim \mathbf{w}}[W(\mathbf{s})]$. We want to lower bound how far the average welfare of the sequence is, with respect to the optimal welfare of the static game:

$$\text{OPT} = \max_{\mathbf{s} \in S_1 \times \ldots \times S_n} W(\mathbf{s}).$$

This is the optimal welfare achievable in the absence of player incentives and if a central coordinator could dictate each player's strategy. We next define a class of games first identified by Roughgarden [18] on which we can approximate the optimal welfare using decoupled no-regret dynamics.

**Definition 1** (Smooth game [18])**.** *A game is $(\lambda, \mu)$-smooth if there exists a strategy profile $\mathbf{s}^*$ such that for any strategy profile $\mathbf{s}$: $\sum_{i \in N} u_i(s_i^*, \mathbf{s}_{-i}) \geq \lambda \text{OPT} - \mu W(\mathbf{s})$.*

In words, any player using his optimal strategy continues to do well irrespective of other players' strategies. This condition directly implies near-optimality of no-regret dynamics as we show below.

**Proposition 2.** *In a $(\lambda, \mu)$-smooth game, if each player $i$ suffers regret at most $r_i(T)$, then:*

$$\frac{1}{T} \sum_{t=1}^{T} W(\mathbf{w}^t) \geq \frac{\lambda}{1+\mu} \text{OPT} - \frac{1}{1+\mu} \frac{1}{T} \sum_{i \in N} r_i(T) = \frac{1}{\rho} \text{OPT} - \frac{1}{1+\mu} \frac{1}{T} \sum_{i \in N} r_i(T),$$

*where the factor $\rho = (1+\mu)/\lambda$ is called the* price of anarchy *(POA).*

This proposition is essentially a more explicit version of Roughgarden's result [18]; we provide a proof in the appendix for completeness. The result shows that the convergence to POA is driven by the quantity $\frac{1}{1+\mu} \frac{1}{T} \sum_{i \in N} r_i(T)$. There are many algorithms which achieve a regret rate of $r_i(T) = O(\sqrt{\log(d)T})$, in which case the latter theorem would imply that the average welfare converges to POA at a rate of $O(n\sqrt{\log(d)/T})$. As we will show, for some natural classes of no-regret algorithms the average welfare converges at the much faster rate of $O(n^2 \log(d)/T)$.

## 3   Fast Convergence to Approximate Efficiency

In this section, we present our main theoretical results characterizing a class of no-regret dynamics which lead to faster convergence in smooth games. We begin by describing this class.

**Definition 3** (RVU property)**.** *We say that a vanishing regret algorithm satisfies the Regret bounded by Variation in Utilities (RVU) property with parameters $\alpha > 0$ and $0 < \beta \leq \gamma$ and a pair of dual norms $(\|\cdot\|, \|\cdot\|_*)$[1] if its regret on any sequence of utilities $\mathbf{u}^1, \mathbf{u}^2, \ldots, \mathbf{u}^T$ is bounded as*

$$\sum_{t=1}^{T} \langle \mathbf{w}^* - \mathbf{w}^t, \mathbf{u}^t \rangle \leq \alpha + \beta \sum_{t=1}^{T} \|\mathbf{u}^t - \mathbf{u}^{t-1}\|_*^2 - \gamma \sum_{t=1}^{T} \|\mathbf{w}^t - \mathbf{w}^{t-1}\|^2. \tag{1}$$

Typical online learning algorithms such as Mirror Descent and FTRL do not satisfy the RVU property in their vanilla form, as the middle term grows as $\sum_{t=1}^{T} \|\mathbf{u}^t\|_*^2$ for these methods. However, Rakhlin and Sridharan [16] give a modification of Mirror Descent with this property, and we will present a similar variant of FTRL in the sequel.

We now present two sets of results when each player uses an algorithm with this property. The first discusses the convergence of social welfare, while the second governs the convergence of the individual players' utilities at a fast rate.

## 3.1 Fast Convergence of Social Welfare

Given Proposition 2, we only need to understand the evolution of the sum of players' regrets $\sum_{t=1}^{T} r_i(T)$ in order to obtain convergence rates of the social welfare. Our main result in this section bounds this sum when each player uses dynamics with the RVU property.

**Theorem 4.** *Suppose that the algorithm of each player $i$ satisfies the property RVU with parameters $\alpha, \beta$ and $\gamma$ such that $\beta \leq \gamma/(n-1)^2$ and $\|\cdot\| = \|\cdot\|_1$. Then $\sum_{i \in N} r_i(T) \leq \alpha n$.*

*Proof.* Since $u_i(\mathbf{s}) \leq 1$, definitions imply: $\|\mathbf{u}_i^t - \mathbf{u}_i^{t-1}\|_* \leq \sum_{\mathbf{s}_{-i}} \left| \prod_{j \neq i} w_{j,s_j}^t - \prod_{j \neq i} w_{j,s_j}^{t-1} \right|$. The latter is the total variation distance of two product distributions. By known properties of total variation (see e.g. [11]), this is bounded by the sum of the total variations of each marginal distribution:

$$\sum_{\mathbf{s}_{-i}} \left| \prod_{j \neq i} w_{j,s_j}^t - \prod_{j \neq i} w_{j,s_j}^{t-1} \right| \leq \sum_{j \neq i} \|\mathbf{w}_j^t - \mathbf{w}_j^{t-1}\| \tag{2}$$

By Jensen's inequality, $\left( \sum_{j \neq i} \|\mathbf{w}_j^t - \mathbf{w}_j^{t-1}\| \right)^2 \leq (n-1) \sum_{j \neq i} \|\mathbf{w}_j^t - \mathbf{w}_j^{t-1}\|^2$, so that

$$\sum_{i \in N} \|\mathbf{u}_i^t - \mathbf{u}_i^{t-1}\|_*^2 \leq (n-1) \sum_{i \in N} \sum_{j \neq i} \|\mathbf{w}_j^t - \mathbf{w}_j^{t-1}\|^2 = (n-1)^2 \sum_{i \in N} \|\mathbf{w}_i^t - \mathbf{w}_i^{t-1}\|^2.$$

The theorem follows by summing up the RVU property (1) for each player $i$ and observing that the summation of the second terms is smaller than that of the third terms and thereby can be dropped. ∎

**Remark:** The rates from the theorem depend on $\alpha$, which will be $O(1)$ in the sequel. The above theorem extends to the case where $\|\cdot\|$ is any norm equivalent to the $\ell_1$ norm. The resulting requirement on $\beta$ in terms of $\gamma$ can however be more stringent. Also, the theorem does not require that all players use the same no-regret algorithm unlike previous results [4, 17], as long as each player's algorithm satisfies the RVU property with a common bound on the constants.

We now instantiate the result with examples that satisfy the RVU property with different constants.

### 3.1.1 Optimistic Mirror Descent

The optimistic mirror descent (OMD) algorithm of Rakhlin and Sridharan [16] is parameterized by an adaptive predictor sequence $\mathbf{M}_i^t$ and a regularizer[2] $\mathcal{R}$ which is 1-strongly convex[3] with respect to a norm $\|\cdot\|$. Let $D_{\mathcal{R}}$ denote the Bregman divergence associated with $\mathcal{R}$. Then the update rule is defined as follows: let $\mathbf{g}_i^0 = \operatorname{argmin}_{\mathbf{g} \in \Delta(S_i)} \mathcal{R}(\mathbf{g})$ and

$$\Phi(\mathbf{u}, \mathbf{g}) = \operatorname*{argmax}_{\mathbf{w} \in \Delta(S_i)} \eta \cdot \langle \mathbf{w}, \mathbf{u} \rangle - D_{\mathcal{R}}(\mathbf{w}, \mathbf{g}),$$

then:

$$\mathbf{w}_i^t = \Phi(\mathbf{M}_i^t, \mathbf{g}_i^{t-1}), \quad \text{and} \quad \mathbf{g}_i^t = \Phi(\mathbf{u}_i^t, \mathbf{g}_i^{t-1})$$

Then the following proposition can be obtained for this method.

**Proposition 5.** *The OMD algorithm using stepsize $\eta$ and $\mathbf{M}_i^t = \mathbf{u}_i^{t-1}$ satisfies the RVU property with constants $\alpha = R/\eta$, $\beta = \eta$, $\gamma = 1/(8\eta)$, where $R = \max_i \sup_f D_{\mathcal{R}}(f, \mathbf{g}_i^0)$.*

The proposition follows by further crystallizing the arguments of Rakhlin and Sridaran [17], and we provide a proof in the appendix for completeness. The above proposition, along with Theorem 4, immediately yields the following corollary, which had been proved by Rakhlin and Sridharan [17] for two-person zero-sum games, and which we here extend to general games.

**Corollary 6.** *If each player runs OMD with $\mathbf{M}_i^t = \mathbf{u}_i^{t-1}$ and stepsize $\eta = 1/(\sqrt{8}(n-1))$, then we have $\sum_{i \in N} r_i(T) \leq nR/\eta \leq n(n-1)\sqrt{8}R = O(1)$.*

The corollary follows by noting that the condition $\beta \leq \gamma/(n-1)^2$ is met with our choice of $\eta$.

### 3.1.2 Optimistic Follow the Regularized Leader

We next consider a different class of algorithms denoted as *optimistic follow the regularized leader* (*OFTRL*). This algorithm is similar but not equivalent to OMD, and is an analogous extension of standard FTRL [12]. This algorithm takes the same parameters as for OMD and is defined as follows: Let $\mathbf{w}_i^0 = \operatorname{argmin}_{\mathbf{w} \in \Delta(S_i)} \mathcal{R}(\mathbf{w})$ and:

$$\mathbf{w}_i^T = \operatorname*{argmax}_{\mathbf{w} \in \Delta(S_i)} \left\langle \mathbf{w}, \sum_{t=1}^{T-1} \mathbf{u}_i^t + \mathbf{M}_i^T \right\rangle - \frac{\mathcal{R}(\mathbf{w})}{\eta}.$$

We consider three variants of OFTRL with different choices of the sequence $\mathbf{M}_i^t$, incorporating the recency bias in different forms.

**One-step recency bias:** The simplest form of OFTRL uses $\mathbf{M}_i^t = \mathbf{u}_i^{t-1}$ and obtains the following result, where $R = \max_i \left( \sup_{\mathbf{f} \in \Delta(S_i)} \mathcal{R}(\mathbf{f}) - \inf_{\mathbf{f} \in \Delta(S_i)} \mathcal{R}(\mathbf{f}) \right)$.

**Proposition 7.** *The OFTRL algorithm using stepsize $\eta$ and $\mathbf{M}_i^t = \mathbf{u}_i^{t-1}$ satisfies the RVU property with constants $\alpha = R/\eta$, $\beta = \eta$ and $\gamma = 1/(4\eta)$.*

Combined with Theorem 4, this yields the following constant bound on the total regret of all players:

**Corollary 8.** *If each player runs OFTRL with $\mathbf{M}_i^t = \mathbf{u}_i^{t-1}$ and $\eta = 1/(2(n-1))$, then we have $\sum_{i \in N} r_i(T) \leq nR/\eta \leq 2n(n-1)R = O(1)$.*

Rakhlin and Sridharan [16] also analyze an FTRL variant, but require a self-concordant barrier for the constraint set as opposed to an arbitrary strongly convex regularizer, and their bound is missing the crucial negative terms of the RVU property which are essential for obtaining Theorem 4.

**$H$-step recency bias:** More generally, given a window size $H$, one can define $\mathbf{M}_i^t = \sum_{\tau=t-H}^{t-1} \mathbf{u}_i^\tau / H$. We have the following proposition.

**Proposition 9.** *The OFTRL algorithm using stepsize $\eta$ and $\mathbf{M}_i^t = \sum_{\tau=t-H}^{t-1} \mathbf{u}_i^\tau / H$ satisfies the RVU property with constants $\alpha = R/\eta$, $\beta = \eta H^2$ and $\gamma = 1/(4\eta)$.*

Setting $\eta = 1/(2H(n-1))$, we obtain the analogue of Corollary 8, with an extra factor of $H$.

**Geometrically discounted recency bias:** The next proposition considers an alternative form of recency bias which includes all the previous utilities, but with a geometric discounting.

**Proposition 10.** *The OFTRL algorithm using stepsize $\eta$ and $\mathbf{M}_i^t = \frac{1}{\sum_{\tau=0}^{t-1} \delta^{-\tau}} \sum_{\tau=0}^{t-1} \delta^{-\tau} \mathbf{u}_i^\tau$ satisfies the RVU property with constants $\alpha = R/\eta$, $\beta = \eta/(1-\delta)^3$ and $\gamma = 1/(8\eta)$.*

Note that these choices for $\mathbf{M}_i^t$ can also be used in OMD with qualitatively similar results.

## 3.2 Fast Convergence of Individual Utilities

The previous section shows implications of the RVU property on the social welfare. This section complements these with a similar result for each player's individual utility.

**Theorem 11.** *Suppose that the players use algorithms satisfying the RVU property with parameters $\alpha > 0, \beta > 0, \gamma \geq 0$. If we further have the stability property $\|\mathbf{w}_i^t - \mathbf{w}_i^{t+1}\| \leq \kappa$, then for any player $\sum_{t=1}^{T} \langle \mathbf{w}_i^* - \mathbf{w}_i^t, \mathbf{u}_i^t \rangle \leq \alpha + \beta \kappa^2 (n-1)^2 T$.*

Similar reasoning as in Theorem 4 yields: $\|\mathbf{u}_i^t - \mathbf{u}_i^{t-1}\|_*^2 \leq (n-1) \sum_{j \neq i} \|\mathbf{w}_j^t - \mathbf{w}_j^{t-1}\|^2 \leq (n-1)^2 \kappa^2$, and summing the terms gives the theorem.

Noting that OFTRL satisfies the RVU property with constants given in Proposition 7 and stability property with $\kappa = 2\eta$ (see Lemma 20 in the appendix), we have the following corollary.

**Corollary 12.** *If all players use the OFTRL algorithm with $\mathbf{M}_i^t = \mathbf{u}_i^{t-1}$ and $\eta = (n-1)^{-1/2} T^{-1/4}$, then we have $\sum_{t=1}^{T} \langle \mathbf{w}_i^* - \mathbf{w}_i^t, \mathbf{u}_i^t \rangle \leq (R+4)\sqrt{n-1} \cdot T^{1/4}$.*

Similar results hold for the other forms of recency bias, as well as for OMD. Corollary 12 gives a fast convergence rate of the players' strategies to the set of *coarse correlated equilibria* (CCE) of the game. This improves the previously known convergence rate $\sqrt{T}$ (e.g. [10]) to CCE using natural, decoupled no-regret dynamics defined in [4].

## 4  Robustness to Adversarial Opponent

So far we have shown simple dynamics with rapid convergence properties in favorable environments when each player in the game uses an algorithm with the RVU property. It is natural to wonder if this comes at the cost of worst-case guarantees when some players do not use algorithms with this property. Rakhlin and Sridharan [17] address this concern by modifying the OMD algorithm with additional smoothing and adaptive step-sizes so as to preserve the fast rates in the favorable case while still guaranteeing $O(1/\sqrt{T})$ regret for each player, no matter how the opponents play. It is not so obvious how this modification might extend to other procedures, and it seems undesirable to abandon the black-box regret transformations we used to obtain Theorem 4. In this section, we present a generic way of transforming an algorithm which satisfies the RVU property so that it retains the fast convergence in favorable settings, but always guarantees a worst-case regret of $\tilde{O}(1/\sqrt{T})$.

In order to present our modification, we need a parametric form of the RVU property which will also involve a tunable parameter of the algorithm. For most online learning algorithms, this will correspond to the step-size parameter used by the algorithm.

**Definition 13** (RVU($\rho$) property). *We say that a parametric algorithm $\mathcal{A}(\rho)$ satisfies the Regret bounded by Variation in Utilities($\rho$) (RVU($\rho$)) property with parameters $\alpha, \beta, \gamma > 0$ and a pair of dual norms $(\|\cdot\|, \|\cdot\|_*)$ if its regret on any sequence of utilities $\mathbf{u}^1, \mathbf{u}^2, \ldots, \mathbf{u}^T$ is bounded as*

$$\sum_{t=1}^{T} \left\langle \mathbf{w}^* - \mathbf{w}^t, \mathbf{u}^t \right\rangle \leq \frac{\alpha}{\rho} + \rho\beta \sum_{t=1}^{T} \|\mathbf{u}^t - \mathbf{u}^{t-1}\|_*^2 - \frac{\gamma}{\rho} \sum_{t=1}^{T} \|\mathbf{w}^t - \mathbf{w}^{t-1}\|^2. \tag{3}$$

In both OMD and OFTRL algorithms from Section 3, the parameter $\rho$ is precisely the stepsize $\eta$. We now show an adaptive choice of $\rho$ according to an epoch-based doubling schedule.

**Black-box reduction.**   Given a parametric algorithm $\mathcal{A}(\rho)$ as a black-box we construct a wrapper $\mathcal{A}'$ based on the doubling trick: The algorithm of each player proceeds in epochs. At each epoch $r$ the player $i$ has an upper bound of $B_r$ on the quantity $\sum_{t=1}^{T} \|\mathbf{u}_i^t - \mathbf{u}_i^{t-1}\|_*^2$. We start with a parameter $\eta_*$ and $B_1 = 1$, and for $\tau = 1, 2, \ldots, T$ repeat:

1. Play according to $\mathcal{A}(\eta_r)$ and receive $\mathbf{u}_i^\tau$.
2. If $\sum_{t=1}^{\tau} |\mathbf{u}_i^t - \mathbf{u}_i^{t-1}\|_*^2 \geq B_r$:
   (a) Update $r \leftarrow r + 1$, $B_r \leftarrow 2B_r$, $\eta_r = \min\left\{\frac{\alpha}{\sqrt{B_r}}, \eta_*\right\}$, with $\alpha$ as in Equation (3).
   (b) Start a new run of $\mathcal{A}$ with parameter $\eta_r$.

**Theorem 14.** *Algorithm $\mathcal{A}'$ achieves regret at most the minimum of the following two terms:*

$$\sum_{t=1}^{T} \left\langle \mathbf{w}_i^* - \mathbf{w}_i^t, \mathbf{u}_i^t \right\rangle \leq \log(T)\left(2 + \frac{\alpha}{\eta_*} + (2 + \eta_* \cdot \beta) \sum_{t=1}^{T} \|\mathbf{u}_i^t - \mathbf{u}_i^{t-1}\|_*^2\right) - \frac{\gamma}{\eta_*} \sum_{t=1}^{T} \|\mathbf{w}_i^t - \mathbf{w}_i^{t-1}\|^2; \tag{4}$$

$$\sum_{t=1}^{T} \left\langle \mathbf{w}_i^* - \mathbf{w}_i^t, \mathbf{u}_i^t \right\rangle \leq \log(T)\left(1 + \frac{\alpha}{\eta_*} + (1 + \alpha \cdot \beta) \cdot \sqrt{2\sum_{t=1}^{T} \|\mathbf{u}_i^t - \mathbf{u}_i^{t-1}\|_*^2}\right) \tag{5}$$

That is, the algorithm satisfies the RVU property, and also has regret that can never exceed $\tilde{O}(\sqrt{T})$. The theorem thus yields the following corollary, which illustrates the stated robustness of $\mathcal{A}'$.

**Corollary 15.** *Algorithm $\mathcal{A}'$, with $\eta_* = \frac{\gamma}{(2+\beta)(n-1)^2 \log(T)}$, achieves regret $\tilde{O}(\sqrt{T})$ against any adversarial sequence, while at the same time satisfying the conditions of Theorem 4. Thereby, if all players use such an algorithm, then: $\sum_{i \in N} r_i(T) \leq n \log(T)(\alpha/\eta_* + 2) = \tilde{O}(1)$.*

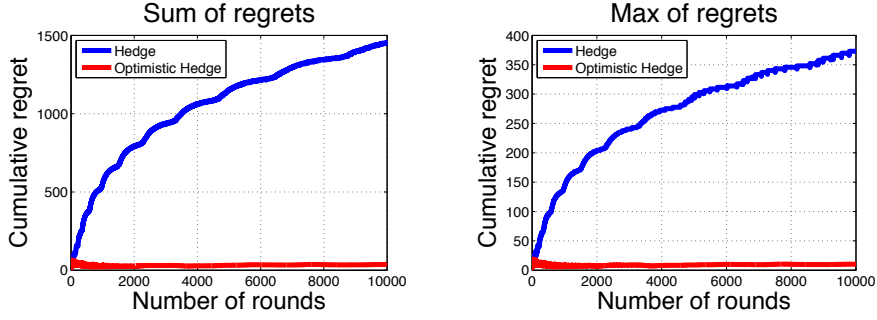

Figure 1: Maximum and sum of individual regrets over time under the Hedge (blue) and Optimistic Hedge (red) dynamics.

*Proof.* Observe that for such $\eta^*$, we have that: $(2 + \eta_* \cdot \beta) \log(T) \leq (2 + \beta) \log(T) \leq \frac{\gamma}{\eta_*(n-1)^2}$. Therefore, algorithm $\mathcal{A}'$, satisfies the sufficient conditions of Theorem 4. ∎

If $\mathcal{A}(\rho)$ is the OFTRL algorithm, then we know by Proposition 7 that the above result applies with $\alpha = R = \max_{\mathbf{w}} \mathcal{R}(\mathbf{w})$, $\beta = 1$, $\gamma = \frac{1}{4}$ and $\rho = \eta$. Setting $\eta_* = \frac{\gamma}{(2+\beta)(n-1)^2} = \frac{1}{12(n-1)^2}$, the resulting algorithm $\mathcal{A}'$ will have regret at most: $\tilde{O}(n^2\sqrt{T})$ against an arbitrary adversary, while if all players use algorithm $\mathcal{A}'$ then $\sum_{i \in N} r_i(T) = O(n^3 \log(T))$.

An analogue of Theorem 11 can also be established for this algorithm:

**Corollary 16.** *If $\mathcal{A}$ satisfies the RVU($\rho$) property, and also $\|\mathbf{w}_i^t - \mathbf{w}_i^{t-1}\| \leq \kappa\rho$, then $\mathcal{A}'$ with $\eta_* = T^{-1/4}$ achieves regret $\tilde{O}(T^{1/4})$ if played against itself, and $\tilde{O}(\sqrt{T})$ against any opponent.*

Once again, OFTRL satisfies the above conditions with $\kappa = 2$, implying robust convergence.

## 5 Experimental Evaluation

We analyzed the performance of optimistic follow the regularized leader with the entropy regularizer, which corresponds to the Hedge algorithm [7] modified so that the last iteration's utility for each strategy is double counted; we refer to it as *Optimistic Hedge*. More formally, the probability of player $i$ playing strategy $j$ at iteration $T$ is proportional to $\exp\left(-\eta \cdot \left(\sum_{t=1}^{T-2} u_{ij}^t + 2u_{ij}^{T-1}\right)\right)$, rather than $\exp\left(-\eta \cdot \sum_{t=1}^{T-1} u_{ij}^t\right)$ as is standard for Hedge.

We studied a simple auction where $n$ players are bidding for $m$ items. Each player has a value $v$ for getting at least one item and no extra value for more items. The utility of a player is the value for the allocation he derived minus the payment he has to make. The game is defined as follows: simultaneously each player picks one of the $m$ items and submits a bid on that item (we assume bids to be discretized). For each item, the highest bidder wins and pays his bid. We let players play this game repeatedly with each player invoking either Hedge or optimistic Hedge. This game, and generalizations of it, are known to be $(1 - 1/e, 0)$-smooth [20], if we also view the auctioneer as a player whose utility is the revenue. The welfare of the game is the value of the resulting allocation, hence not a constant-sum game. The welfare maximization problem corresponds to the unweighted bipartite matching problem. The POA captures how far from the optimal matching is the average allocation of the dynamics. By smoothness we know it converges to at least $1 - 1/e$ of the optimal.

**Fast convergence of individual and average regret.** We run the game for $n = 4$ bidders and $m = 4$ items and valuation $v = 20$. The bids are discretized to be any integer in $[1, 20]$. We find that the sum of the regrets and the maximum individual regret of each player are remarkably lower under Optimistic Hedge as opposed to Hedge. In Figure 1 we plot the maximum individual regret as well as the sum of the regrets under the two algorithms, using $\eta = 0.1$ for both methods. Thus convergence to the set of coarse correlated equilibria is substantially faster under Optimistic Hedge,

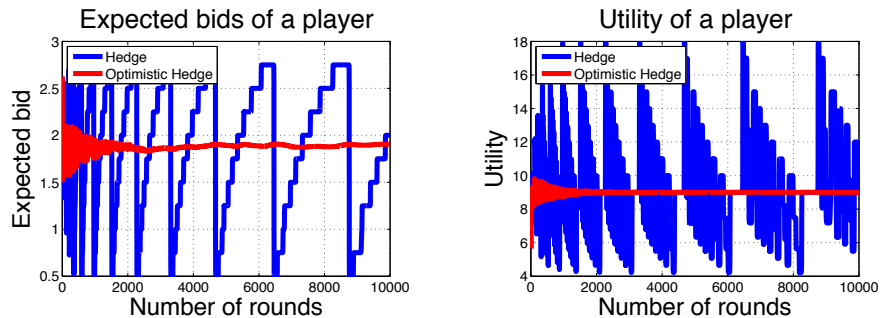

Figure 2: Expected bid and per-iteration utility of a player on one of the four items over time, under Hedge (blue) and Optimistic Hedge (red) dynamics.

confirming our results in Section 3.2. We also observe similar behavior when each player only has value on a randomly picked player-specific subset of items, or uses other step sizes.

**More stable dynamics.** We observe that the behavior under Optimistic Hedge is more stable than under Hedge. In Figure 2, we plot the expected bid of a player on one of the items and his expected utility under the two dynamics. Hedge exhibits the sawtooth behavior that was observed in generalized first price auction run by Overture (see [5, p. 21]). In stunning contrast, Optimistic Hedge leads to more stable expected bids over time. This stability property of optimistic Hedge is one of the main intuitive reasons for the fast convergence of its regret.

**Welfare.** In this class of games, we did not observe any significant difference between the average welfare of the methods. The key reason is the following: the proof that no-regret dynamics are approximately efficient (Proposition 2) only relies on the fact that each player does not have regret against the strategy $s_i^*$ used in the definition of a smooth game. In this game, regret against these strategies is experimentally comparable under both algorithms, even though regret against the best fixed strategy is remarkably different. This indicates a possibility for faster rates for Hedge in terms of welfare. In Appendix H, we show fast convergence of the efficiency of Hedge for cost-minimization games, though with a worse POA .

# 6  Discussion

This work extends and generalizes a growing body of work on decentralized no-regret dynamics in many ways. We demonstrate a class of no-regret algorithms which enjoy rapid convergence when played against each other, while being robust to adversarial opponents. This has implications in computation of correlated equilibria, as well as understanding the behavior of agents in complex multi-player games. There are a number of interesting questions and directions for future research which are suggested by our results, including the following:

**Convergence rates for vanilla Hedge:** The fast rates of our paper do not apply to algorithms such as Hedge without modification. Is this modification to satisfy RVU only sufficient or also necessary? If not, are there counterexamples? In the supplement, we include a sketch hinting at such a counterexample, but also showing fast rates to a worse equilibrium than our optimistic algorithms.

**Convergence of players' strategies:** The OFTRL algorithm often produces much more stable trajectories empirically, as the players converge to an equilibrium, as opposed to say Hedge. A precise quantification of this desirable behavior would be of great interest.

**Better rates with partial information:** If the players do not observe the expected utility function, but only the moves of the other players at each round, can we still obtain faster rates?

## Footnotes

[1]The dual to a norm $\|\cdot\|$ is defined as $\|v\|_* = \sup_{\|u\| \leq 1} \langle u, v \rangle$.

[2] Here and in the sequel, we can use a different regularizer $\mathcal{R}_i$ for each player $i$, without qualitatively affecting any of the results.

[3] $\mathcal{R}$ is 1-strongly convex if $\mathcal{R}\left(\frac{u+v}{2}\right) \leq \frac{\mathcal{R}(u)+\mathcal{R}(v)}{2} - \frac{\|u-v\|^2}{8}, \forall u, v$.

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
