[Supplementary Material]

# Supplementary material for "Fast Convergence of Regularized Learning in Games"

## A  Proof of Proposition 2

**Proposition 2.**  *In a $(\lambda, \mu)$-smooth game, if each player $i$ suffers regret at most $r_i(T)$, then:*

$$\frac{1}{T}\sum_{t=1}^{T}W(\mathbf{w}^t) \geq \frac{\lambda}{1+\mu}\mathrm{OPT} - \frac{1}{1+\mu}\frac{1}{T}\sum_{i\in N}r_i(T) = \frac{1}{\rho}\mathrm{OPT} - \frac{1}{1+\mu}\frac{1}{T}\sum_{i\in N}r_i(T),$$

*where the factor $\rho = (1+\mu)/\lambda$ is called the* price of total anarchy *(POA).*

*Proof.*  Since each player $i$ has regret $r_i(T)$, we have that:

$$\sum_{t=1}^{T}\left\langle \mathbf{w}_i^t, \mathbf{u}_i^t \right\rangle \geq \sum_{t=1}^{T}u_{i,s_i^*}^t - r_i(T) \tag{6}$$

Summing over all players and using the smoothness property:

$$\begin{aligned}
\sum_{t=1}^{T}W(\mathbf{w}^t) &= \sum_{t=1}^{T}\sum_{i\in N}\left\langle \mathbf{w}_i^t, \mathbf{u}_i^t \right\rangle \geq \sum_{t=1}^{T}\sum_{i\in N}u_{i,s_i^*}^t - \sum_{i\in N}r_i(T) \\
&= \sum_{t=1}^{T}\mathbb{E}_{\mathbf{s}\sim\mathbf{w}^t}\left[\sum_{i\in N}u_i(s_i^*, \mathbf{s}_{-i})\right] - \sum_{i\in N}r_i(T) \\
&\geq \sum_{t=1}^{T}\left(\lambda\mathrm{OPT} - \mu E_{\mathbf{s}\sim\mathbf{w}^t}\left[W(\mathbf{s})\right]\right) - \sum_{i\in N}r_i(T) \\
&= \sum_{t=1}^{T}\left(\lambda\mathrm{OPT} - \mu W(\mathbf{w}^t)\right) - \sum_{i\in N}r_i(T)
\end{aligned}$$

By re-arranging we get the result.  ∎

## B  Proof of Proposition 5

**Proposition 5.**  *The OMD algorithm using stepsize $\eta$ and $\mathbf{M}_i^t = \mathbf{u}_i^{t-1}$ satisfies the RVU property with constants $\alpha = R/\eta$, $\beta = \eta$, $\gamma = 1/(8\eta)$, where $R = \max_i \sup_f D_{\mathcal{R}}(f, \mathbf{g}_i^0)$.*

We will use the following theorem of [17].

**Theorem 17** (Raklin and Sridharan [17])**.**  *The regret of a player under optimistic mirror descent and with respect to any $\mathbf{w}_i^* \in \Delta(S_i)$ is upper bounded by:*

$$\sum_{t=1}^{T}\left\langle \mathbf{w}_i^* - \mathbf{w}_i^t, \mathbf{u}_i^t \right\rangle \leq \frac{R}{\eta} + \sum_{t=1}^{T}\|\mathbf{u}_i^t - \mathbf{M}_i^t\|_*\|\mathbf{w}_i^t - \mathbf{g}_i^t\| - \frac{1}{2\eta}\sum_{t=1}^{T}\left(\|\mathbf{w}_i^t - \mathbf{g}_i^t\|^2 + \|\mathbf{w}_i^t - \mathbf{g}_i^{t-1}\|^2\right) \tag{7}$$

*where $R = \sup_f D_{\mathcal{R}}(f, g_0)$.*

We show that if the players use optimistic mirror descent with $\mathbf{M}_i^t = \mathbf{u}_i^{t-1}$, then the regret of each player satisfies the sufficient condition presented in the previous section. Some of the key facts (Equations (9) and (10)) that we use in the following proof appear in [17]. However, the formulation of the regret that we present in the following theorem is not immediately clear in their proof, so we present it here for clarity and completeness.

**Theorem 18.** *The regret of a player under optimistic mirror descent with $\mathbf{M}_i^t = \mathbf{u}_i^{t-1}$ and with respect to any $\mathbf{w}_i^* \in \Delta(S_i)$ is upper bounded by:*

$$\sum_{t=1}^{T} \langle \mathbf{w}_i^* - \mathbf{w}_i^t, \mathbf{u}_i^t \rangle \leq \frac{R}{\eta} + \eta \sum_{t=1}^{T} \|\mathbf{u}_i^t - \mathbf{u}_i^{t-1}\|_*^2 - \frac{1}{8\eta} \sum_{t=1}^{T} \|\mathbf{w}_i^t - \mathbf{w}_i^{t-1}\|^2 \tag{8}$$

*Proof.* By Theorem 17, instantiated for $\mathbf{M}_i^t = \mathbf{u}_i^{t-1}$, we get:

$$\sum_{t=1}^{T} \langle \mathbf{w}_i^* - \mathbf{w}_i^t, \mathbf{u}_i^t \rangle \leq \frac{R}{\eta} + \sum_{t=1}^{T} \|\mathbf{u}_i^t - \mathbf{u}_i^{t-1}\|_* \|\mathbf{w}_i^t - \mathbf{g}_i^t\|$$

$$- \frac{1}{2\eta} \sum_{t=1}^{T} \left( \|\mathbf{w}_i^t - \mathbf{g}_i^t\|^2 + \|\mathbf{w}_i^t - \mathbf{g}_i^{t-1}\|^2 \right)$$

Using the fact that for any $\rho > 0$:

$$\|\mathbf{u}_i^t - \mathbf{M}_i^t\|_* \|\mathbf{w}_i^t - \mathbf{g}_i^t\| \leq \frac{\rho}{2} \|\mathbf{u}_i^t - \mathbf{M}_i^t\|_*^2 + \frac{1}{2\rho} \|\mathbf{w}_i^t - \mathbf{g}_i^t\|^2 \tag{9}$$

We get:

$$\sum_{t=1}^{T} \langle \mathbf{w}_i^* - \mathbf{w}_i^t, \mathbf{u}_i^t \rangle \leq \frac{R}{\eta} + \frac{\rho}{2} \sum_{t=1}^{T} \|\mathbf{u}_i^t - \mathbf{u}_i^{t-1}\|_*^2 - \left( \frac{1}{2\eta} - \frac{1}{2\rho} \right) \sum_{t=1}^{T} \|\mathbf{w}_i^t - \mathbf{g}_i^t\|^2 - \frac{1}{2\eta} \sum_{t=1}^{T} \|\mathbf{w}_i^t - \mathbf{g}_i^{t-1}\|^2$$

For $\rho = 2\eta$, the latter simplifies to:

$$\sum_{t=1}^{T} \langle \mathbf{w}_i^* - \mathbf{w}_i^t, \mathbf{u}_i^t \rangle \leq \frac{R}{\eta} + \eta \sum_{t=1}^{T} \|\mathbf{u}_i^t - \mathbf{u}_i^{t-1}\|_*^2 - \frac{1}{4\eta} \sum_{t=1}^{T} \|\mathbf{w}_i^t - \mathbf{g}_i^t\|^2 - \frac{1}{2\eta} \sum_{t=1}^{T} \|\mathbf{w}_i^t - \mathbf{g}_i^{t-1}\|^2$$

$$\leq \frac{R}{\eta} + \eta \sum_{t=1}^{T} \|\mathbf{u}_i^t - \mathbf{u}_i^{t-1}\|_*^2 - \frac{1}{4\eta} \sum_{t=1}^{T} \|\mathbf{w}_i^t - \mathbf{g}_i^t\|^2 - \frac{1}{4\eta} \sum_{t=1}^{T} \|\mathbf{w}_i^t - \mathbf{g}_i^{t-1}\|^2$$

Last we use the fact that:

$$\|\mathbf{w}_i^t - \mathbf{w}_i^{t-1}\|^2 \leq 2\|\mathbf{w}_i^t - \mathbf{g}_i^{t-1}\|^2 + 2\|\mathbf{w}_i^{t-1} - \mathbf{g}_i^{t-1}\|^2 \tag{10}$$

Summing over all timesteps:

$$\sum_{t=1}^{T} \|\mathbf{w}_i^t - \mathbf{w}_i^{t-1}\|^2 \leq 2\sum_{t=1}^{T} \|\mathbf{w}_i^t - \mathbf{g}_i^{t-1}\|^2 + 2\sum_{t=1}^{T} \|\mathbf{w}_i^{t-1} - \mathbf{g}_i^{t-1}\|^2$$

$$\leq 2\sum_{t=1}^{T} \|\mathbf{w}_i^t - \mathbf{g}_i^{t-1}\|^2 + 2\sum_{t=1}^{T} \|\mathbf{w}_i^t - g_i^t\|^2$$

Dividing over by $\frac{1}{8\eta}$ and applying it in the previous upper bound on the regret, we get:

$$\sum_{t=1}^{T} \langle \mathbf{w}_i^* - \mathbf{w}_i^t, \mathbf{u}_i^t \rangle \leq \frac{R}{\eta} + \eta \sum_{t=1}^{T} \|\mathbf{u}_i^t - \mathbf{u}_i^{t-1}\|_*^2 - \frac{1}{8\eta} \sum_{t=1}^{T} \|\mathbf{w}_i^t - \mathbf{w}_i^{t-1}\|^2$$

∎

## C  Proof of Proposition 7

**Proposition 7.** *The OFTRL algorithm using stepsize $\eta$ and $\mathbf{M}_i^t = \mathbf{u}_i^{t-1}$ satisfies the RVU property with constants $\alpha = R/\eta$, $\beta = \eta$ and $\gamma = 1/(4\eta)$.*

We first show that these algorithms achieve the same regret bounds as optimistic mirror descent. This result does not appear in previous work in any form.

Even though the algorithms do not make use of a secondary sequence, we will still use in the analysis the notation:

$$\mathbf{g}_i^T = \operatorname*{argmax}_{\mathbf{g}\in\Delta(S_i)} \left\langle \mathbf{g}, \sum_{t=1}^{T} \mathbf{u}_i^t \right\rangle - \frac{\mathcal{R}(\mathbf{g})}{\eta}.$$

These secondary variables are often called *be the leader* sequence as they can see one step in the future.

**Theorem 19.** *The regret of a player under optimistic FTRL and with respect to any* $\mathbf{w}_i^* \in \Delta(S_i)$ *is upper bounded by:*

$$\sum_{t=1}^{T} \left\langle \mathbf{w}_i^* - \mathbf{w}_i^t, \mathbf{u}_i^t \right\rangle \leq \frac{R}{\eta} + \sum_{t=1}^{T} \|\mathbf{u}_i^t - \mathbf{M}_i^t\|_* \|\mathbf{w}_i^t - \mathbf{g}_i^t\| - \frac{1}{2\eta}\sum_{t=1}^{T}\left(\|\mathbf{w}_i^t - \mathbf{g}_i^t\|^2 + \|\mathbf{w}_i^t - \mathbf{g}_i^{t-1}\|^2\right)$$

$$(11)$$

*where* $R = \sup_{\mathbf{f}} \mathcal{R}(\mathbf{f}) - \inf_{\mathbf{f}} \mathcal{R}(\mathbf{f})$.

*Proof.* First observe that:

$$\left\langle \mathbf{w}_i^* - \mathbf{w}_i^t, \mathbf{u}_i^t \right\rangle = \left\langle \mathbf{g}_i^t - \mathbf{w}_i^t, \mathbf{u}_i^t - \mathbf{M}_i^t \right\rangle + \left\langle \mathbf{g}_i^t - \mathbf{w}_i^t, \mathbf{M}_i^t \right\rangle + \left\langle \mathbf{w}_i^* - \mathbf{g}_i^t, \mathbf{u}_i^t \right\rangle \qquad (12)$$

Without loss of generality we will assume that $\inf_{\mathbf{f}} \mathcal{R}(\mathbf{f}) = 0$. Since $\langle \mathbf{g}_i^t - \mathbf{w}_i^t, \mathbf{u}_i^t - \mathbf{M}_i^t \rangle \leq \|\mathbf{g}_i^t - \mathbf{w}_i^t\|\|\mathbf{u}_i^t - \mathbf{M}_i^t\|_*$, it suffices to show that for any $\mathbf{w}_i^* \in \Delta(S_i)$:

$$\sum_{t=1}^{T}\left(\left\langle \mathbf{g}_i^t - \mathbf{w}_i^t, \mathbf{M}_i^t \right\rangle + \left\langle \mathbf{w}_i^* - \mathbf{g}_i^t, \mathbf{u}_i^t \right\rangle\right) \leq \frac{\mathcal{R}(\mathbf{w}_i^*)}{\eta} - \frac{1}{2\eta}\sum_{t=1}^{T}\left(\|\mathbf{w}_i^t - \mathbf{g}_i^t\|^2 + \|\mathbf{w}_i^t - \mathbf{g}_i^{t-1}\|^2\right) \quad (13)$$

For shorthand notation let: $I_T = \frac{1}{2\eta}\sum_{t=1}^{T}\left(\|\mathbf{w}_i^t - \mathbf{g}_i^t\|^2 + \|\mathbf{w}_i^t - \mathbf{g}_i^{t-1}\|^2\right)$. By induction assume that for all $\mathbf{w}_i^*$:

$$\sum_{t=1}^{T-1}\left(\left\langle \mathbf{g}_i^t - \mathbf{w}_i^t, \mathbf{M}_i^t \right\rangle - \left\langle \mathbf{g}_i^t, \mathbf{u}_i^t \right\rangle\right) \leq -\sum_{t=1}^{T-1}\left\langle \mathbf{w}_i^*, \mathbf{u}_i^t \right\rangle + \frac{\mathcal{R}(\mathbf{w}_i^*)}{\eta} - I_{T-1}$$

$$= -\left\langle \mathbf{w}_i^*, \sum_{t=1}^{T-1} \mathbf{u}_i^t \right\rangle + \frac{\mathcal{R}(\mathbf{w}_i^*)}{\eta} - I_{T-1}$$

Apply the above for $\mathbf{w}_i^* = \mathbf{g}_i^{T-1}$ and add $\left\langle \mathbf{g}_i^T - \mathbf{w}_i^T, \mathbf{M}_i^T \right\rangle - \left\langle \mathbf{g}_i^T, \mathbf{u}_i^T \right\rangle$ on both sides:

$$\sum_{t=1}^{T}\left(\left\langle \mathbf{g}_i^t - \mathbf{w}_i^t, \mathbf{M}_i^t \right\rangle - \left\langle \mathbf{g}_i^t, \mathbf{u}_i^t \right\rangle\right) \leq -\left\langle \mathbf{g}_i^{T-1}, \sum_{t=1}^{T-1} \mathbf{u}_i^t \right\rangle + \frac{\mathcal{R}(\mathbf{g}_i^{T-1})}{\eta} - I_{T-1} + \left\langle \mathbf{g}_i^T - \mathbf{w}_i^T, \mathbf{M}_i^T \right\rangle - \left\langle \mathbf{g}_i^T, \mathbf{u}_i^T \right\rangle$$

$$\leq -\left\langle \mathbf{w}_i^T, \sum_{t=1}^{T-1} \mathbf{u}_i^t \right\rangle + \frac{\mathcal{R}(\mathbf{w}_i^T)}{\eta} - I_{T-1} + \left\langle \mathbf{g}_i^T - \mathbf{w}_i^T, \mathbf{M}_i^T \right\rangle - \left\langle \mathbf{g}_i^T, \mathbf{u}_i^T \right\rangle$$

$$- \frac{1}{2\eta}\|\mathbf{w}_i^T - \mathbf{g}_i^{T-1}\|^2$$

$$= -\left\langle \mathbf{w}_i^T, \sum_{t=1}^{T-1} \mathbf{u}_i^t + \mathbf{M}_i^T \right\rangle + \frac{\mathcal{R}(\mathbf{w}_i^T)}{\eta} - I_{T-1} + \left\langle \mathbf{g}_i^T, \mathbf{M}_i^T \right\rangle - \left\langle \mathbf{g}_i^T, \mathbf{u}_i^T \right\rangle$$

$$- \frac{1}{2\eta}\|\mathbf{w}_i^T - \mathbf{g}_i^{T-1}\|^2$$

$$\leq -\left\langle \mathbf{g}_i^T, \sum_{t=1}^{T-1} \mathbf{u}_i^t + \mathbf{M}_i^T \right\rangle + \frac{\mathcal{R}(\mathbf{g}_i^T)}{\eta} - I_{T-1} + \left\langle \mathbf{g}_i^T, \mathbf{M}_i^T \right\rangle - \left\langle \mathbf{g}_i^T, \mathbf{u}_i^T \right\rangle$$

$$- \frac{1}{2\eta}\|\mathbf{w}_i^T - \mathbf{g}_i^{T-1}\|^2 - \frac{1}{2\eta}\|\mathbf{w}_i^T - \mathbf{g}_i^T\|^2$$

$$= -\left\langle \mathbf{g}_i^T, \sum_{t=1}^{T} \mathbf{u}_i^t \right\rangle + \frac{\mathcal{R}(\mathbf{g}_i^T)}{\eta} - I_T$$

$$\leq -\left\langle \mathbf{q}_i^*, \sum_{t=1}^{T} \mathbf{u}_i^t \right\rangle + \frac{\mathcal{R}(\mathbf{q}_i^*)}{\eta} - I_T$$

The inequalities follow by the optimality of the corresponding variable that was changed and by the strong convexity of $\mathcal{R}(\cdot)$. The final vector $\mathbf{q}_i^*$ is an arbitrary vector in $\Delta(S_i)$. The base case of $T = 0$ follows trivially by $\mathcal{R}(\mathbf{f}) \geq 0$ for all $\mathbf{f}$. This concludes the inductive proof. ∎

Thus optimistic FTRL achieves the exact same form of regret presented in Theorem 17 for optimistic mirror descent. Hence, the equivalent versions of Theorem 18 and Corollary 6 hold also for the optimistic FTRL algorithm. In fact we are able to show slightly stronger bounds for optimistic FTRL, based on the following lemmas.

**Lemma 20** (Stability). *For the optimistic FTRL algorithm:*

$$\|\mathbf{w}_i^t - \mathbf{g}_i^t\| \leq \eta \cdot \|\mathbf{M}_i^t - \mathbf{u}_i^t\|_* \tag{14}$$

$$\|\mathbf{g}_i^t - \mathbf{w}_i^{t+1}\| \leq \eta \cdot \|\mathbf{M}_i^{t+1}\|_* \tag{15}$$

*Proof.* Let $F_T(\mathbf{f}) = \left\langle \mathbf{f}, \sum_{t=1}^{T-1} \mathbf{u}_i^t + \mathbf{M}_i^T \right\rangle - \eta^{-1}\mathcal{R}(\mathbf{f})$ and $G_T(\mathbf{f}) = \left\langle \mathbf{f}, \sum_{t=1}^{T} \mathbf{u}_i^t \right\rangle - \eta^{-1}\mathcal{R}(\mathbf{f})$. Observe that: $F_T(\mathbf{f}) - G_T(\mathbf{f}) = \left\langle \mathbf{f}, \mathbf{M}_i^T - \mathbf{u}_i^T \right\rangle$ and $F_{T+1}(\mathbf{f}) - G_T(\mathbf{f}) = \left\langle \mathbf{f}, \mathbf{M}_i^{T+1} \right\rangle$.

**Part 1** By the optimality of $\mathbf{w}_i^T$ and $\mathbf{g}_i^T$ and the strong convexity of $\mathcal{R}(\cdot)$:

$$F_T(\mathbf{w}_i^T) \geq F_T(\mathbf{g}_i^T) + \frac{1}{2\eta}\|\mathbf{w}_i^T - \mathbf{g}_i^T\|^2$$

$$G_T(\mathbf{g}_i^T) \geq G_T(\mathbf{w}_i^T) + \frac{1}{2\eta}\|\mathbf{w}_i^T - \mathbf{g}_i^T\|^2$$

Adding both inequalities and using the previous observations:

$$\frac{1}{\eta}\|\mathbf{w}_i^T - \mathbf{g}_i^T\|^2 \leq \left\langle \mathbf{w}_i^T - \mathbf{g}_i^T, \mathbf{M}_i^T - \mathbf{u}_i^T \right\rangle \leq \|\mathbf{w}_i^T - \mathbf{g}_i^T\| \cdot \|\mathbf{M}_i^T - \mathbf{u}_i^T\|_*$$

Dividing over by $\|\mathbf{w}_i^T - \mathbf{g}_i^T\|$ gives the first inequality of the lemma.

**Part 2** By the optimality of $\mathbf{g}_i^T$ and $\mathbf{w}_i^{T+1}$ and strong convexity:

$$F_{T+1}(\mathbf{w}_i^{T+1}) \geq F_{T+1}(\mathbf{g}_i^T) + \frac{1}{2\eta}\|\mathbf{w}_i^{T+1} - \mathbf{g}_i^T\|^2$$

$$G_T(\mathbf{g}_i^T) \geq G_T(\mathbf{w}_i^{T+1}) + \frac{1}{2\eta}\|\mathbf{w}_i^{T+1} - \mathbf{g}_i^T\|^2$$

Adding the inequalities:

$$\frac{1}{\eta}\|\mathbf{w}_i^{T+1} - \mathbf{g}_i^T\|^2 \leq \left\langle \mathbf{w}_i^{T+1} - \mathbf{g}_i^T, \mathbf{M}_i^{T+1} \right\rangle \leq \|\mathbf{w}_i^{T+1} - \mathbf{g}_i^T\| \cdot \|\mathbf{M}_i^{T+1}\|_*$$

Dividing over by $\|\mathbf{w}_i^{T+1} - \mathbf{g}_i^T\|$, yields second inequality of the lemma. ∎

Given Theorem 19 and Lemma 20, the proposition immediately follows since

$$\sum_{t=1}^{T} \left\langle \mathbf{w}_i^* - \mathbf{w}_i^t, \mathbf{u}_i^t \right\rangle \leq \frac{R}{\eta} + \eta \sum_{t=1}^{T} \|\mathbf{u}_i^t - \mathbf{M}_i^t\|_*^2 - \frac{1}{2\eta} \sum_{t=1}^{T} \left( \|\mathbf{w}_i^t - \mathbf{g}_i^t\|^2 + \|\mathbf{w}_i^t - \mathbf{g}_i^{t-1}\|^2 \right).$$

Replacing $\mathbf{M}_i^t$ with $\mathbf{u}_i^{t-1}$ and using Inequality (10), yields the result.

# D   Proof of Proposition 9

**Proposition 9.** *The OFTRL algorithm using stepsize $\eta$ and $\mathbf{M}_i^t = \sum_{\tau=t-H}^{t-1} \mathbf{u}_i^\tau / H$ satisfies the RVU property with constants $\alpha = R/\eta$, $\beta = \eta H^2$ and $\gamma = 1/(4\eta)$.*

The proposition is equivalent to the following lemma, which we will state and prove in this appendix.

**Lemma 21.** *For the optimistic FTRL algorithm with* $\mathbf{M}_i^t = \frac{1}{H}\sum_{\tau=t-H}^{t-1}\mathbf{u}_i^\tau$, *the regret is upper bounded by:*

$$\sum_{t=1}^{T}\left\langle\mathbf{w}_i^* - \mathbf{w}_i^t, \mathbf{u}_i^t\right\rangle \le \frac{R}{\eta} + \eta H^2 \sum_{t=1}^{T}\|\mathbf{u}_i^t - \mathbf{u}_i^{t-1}\|_*^2 - \frac{1}{4\eta}\sum_{t=1}^{T}\|\mathbf{w}_i^t - \mathbf{w}_i^{t-1}\|^2 \qquad (16)$$

*where* $R = \sup_{\mathbf{f}}\mathcal{R}(\mathbf{f}) - \inf_{\mathbf{f}}\mathcal{R}(\mathbf{f})$. *Thus we get* $\sum_i r_i(T) \le \frac{nR}{\eta} = 2n(n-1)HR$ *for* $\eta = \frac{1}{2H(n-1)}$.

*Proof.* Similar to Proposition 7, by Theorem 19, Lemma 20 and Inequality (10) we get:

$$\begin{aligned}
\sum_{t=1}^{T}\left\langle\mathbf{w}_i^* - \mathbf{w}_i^t, \mathbf{u}_i^t\right\rangle &\le \frac{R}{\eta} + \eta\sum_{t=1}^{T}\|\mathbf{u}_i^t - \mathbf{M}_i^t\|_*^2 - \frac{1}{4\eta}\sum_{t=1}^{T}\|\mathbf{w}_i^t - \mathbf{w}_i^{t-1}\|^2 \\
&= \frac{R}{\eta} + \eta\sum_{t=1}^{T}\left\|\mathbf{u}_i^t - \frac{1}{H}\sum_{\tau=t-H}^{t-1}\mathbf{u}_i^\tau\right\|_*^2 - \frac{1}{4\eta}\sum_{t=1}^{T}\|\mathbf{w}_i^t - \mathbf{w}_i^{t-1}\|^2 \\
&= \frac{R}{\eta} + \eta\sum_{t=1}^{T}\left(\frac{1}{H}\sum_{\tau=t-H}^{t-1}\|\mathbf{u}_i^t - \mathbf{u}_i^\tau\|_*\right)^2 - \frac{1}{4\eta}\sum_{t=1}^{T}\|\mathbf{w}_i^t - \mathbf{w}_i^{t-1}\|^2
\end{aligned}$$

By triangle inequality:

$$\begin{aligned}
\frac{1}{H}\sum_{\tau=t-H}^{t-1}\|\mathbf{u}_i^t - \mathbf{u}_i^\tau\|_* &\le \frac{1}{H}\sum_{\tau=t-H}^{t-1}\sum_{q=\tau}^{t-1}\left\|\mathbf{u}_i^{q+1} - \mathbf{u}_i^q\right\|_* \\
&= \sum_{\tau=t-H}^{t-1}\frac{t-\tau}{H}\|\mathbf{u}_i^{\tau+1} - \mathbf{u}_i^\tau\|_* \le \sum_{\tau=t-H}^{t-1}\|\mathbf{u}_i^{\tau+1} - \mathbf{u}_i^\tau\|_*
\end{aligned}$$

By Cauchy-Schwarz:

$$\left(\sum_{\tau=t-H}^{t-1}\|\mathbf{u}_i^{\tau+1} - \mathbf{u}_i^\tau\|_*\right)^2 \le H\sum_{\tau=t-H}^{t-1}\|\mathbf{u}_i^{\tau+1} - \mathbf{u}_i^\tau\|_*^2$$

Thus we can derive that:

$$\begin{aligned}
\sum_{t=1}^{T}\left\langle\mathbf{w}_i^* - \mathbf{w}_i^t, \mathbf{u}_i^t\right\rangle &\le \frac{R}{\eta} + \eta H\sum_{t=1}^{T}\sum_{\tau=t-H}^{t-1}\|\mathbf{u}_i^{\tau+1} - \mathbf{u}_i^\tau\|_*^2 - \frac{1}{4\eta}\sum_{t=1}^{T}\|\mathbf{w}_i^t - \mathbf{w}_i^{t-1}\|^2 \\
&\le \frac{R}{\eta} + \eta H^2\sum_{t=1}^{T}\|\mathbf{u}_i^t - \mathbf{u}_i^{t-1}\|_*^2 - \frac{1}{4\eta}\sum_{t=1}^{T}\|\mathbf{w}_i^t - \mathbf{w}_i^{t-1}\|^2
\end{aligned}$$

∎

# E  Proof of Proposition 10

**Proposition 10.** *The OFTRL algorithm using stepsize* $\eta$ *and* $\mathbf{M}_i^t = \frac{1}{\sum_{\tau=0}^{t-1}\delta^{-\tau}}\sum_{\tau=0}^{t-1}\delta^{-\tau}\mathbf{u}_i^\tau$ *satisfies the* RVU *property with constants* $\alpha = R/\eta$, $\beta = \eta/(1-\delta)^3$ *and* $\gamma = 1/(8\eta)$.

The proposition is equivalent to the following lemma which we will prove in this appendix.

**Lemma 22.** *For the optimistic FTRL algorithm with* $\mathbf{M}_i^t = \frac{1}{\sum_{\tau=0}^{t}\delta^{-\tau}}\sum_{\tau=0}^{t-1}\delta^{-\tau}\mathbf{u}_i^\tau$ *for some discount rate* $\delta \in (0,1)$, *the regret is upper bounded by:*

$$\sum_{t=1}^{T}\left\langle\mathbf{w}_i^* - \mathbf{w}_i^t, \mathbf{u}_i^t\right\rangle \le \frac{R}{\eta} + \frac{\eta}{(1-\delta)^3}\sum_{t=1}^{T}\|\mathbf{u}_i^t - \mathbf{u}_i^{t-1}\|_*^2 - \frac{1}{8\eta}\sum_{t=1}^{T}\|\mathbf{w}_i^t - \mathbf{w}_i^{t-1}\|^2 \qquad (17)$$

where $R = \sup_{\mathbf{f}} \mathcal{R}(\mathbf{f}) - \inf_{\mathbf{f}} \mathcal{R}(\mathbf{f})$. *Thus we get* $\sum_i r_i(T) \leq \frac{nR}{\eta} = 2n(n-1)\frac{1}{(1-\delta)^{3/2}}R$ *for* $\eta = \frac{(1-\delta)^{3/2}}{2(n-1)}$.

*Proof.* We show the theorem for the case of optimistic FTRL. The OMD case follows analogously. Similar to Lemma 21 the regret is upper bounded by:

$$
\sum_{t=1}^{T} \left\langle \mathbf{w}_i^* - \mathbf{w}_i^t, \mathbf{u}_i^t \right\rangle \leq \frac{R}{\eta} + \eta \sum_{t=1}^{T} \|\mathbf{u}_i^t - \mathbf{M}_i^t\|_*^2 - \frac{1}{4\eta}\sum_{t=1}^{T} \|\mathbf{w}_i^t - \mathbf{w}_i^{t-1}\|^2
$$

$$
= \frac{R}{\eta} + \eta \sum_{t=1}^{T} \left\| \mathbf{u}_i^t - \frac{1}{\sum_{\tau=0}^{t-1} \delta^{-\tau}} \sum_{\tau=0}^{t-1} \delta^{-\tau} \mathbf{u}_i^\tau \right\|_*^2 - \frac{1}{4\eta}\sum_{t=1}^{T} \|\mathbf{w}_i^t - \mathbf{w}_i^{t-1}\|^2
$$

We will now show that:

$$
\sum_{t=1}^{T} \left\| \mathbf{u}_i^t - \frac{1}{\sum_{\tau=0}^{t-1} \delta^{-\tau}} \sum_{\tau=0}^{t-1} \delta^{-\tau} \mathbf{u}_i^\tau \right\|_*^2 \leq \frac{1}{(1-\delta)^3} \sum_{t=1}^{T} \|\mathbf{u}_i^t - \mathbf{u}_i^{t-1}\|_*^2
$$

which will conclude the proof.

First observe by triangle inequality:

$$
\left\| \mathbf{u}_i^t - \frac{1}{\sum_{\tau=0}^{t-1} \delta^{-\tau}} \sum_{\tau=0}^{t-1} \delta^{-\tau} \mathbf{u}_i^\tau \right\|_* = \frac{1}{\sum_{\tau=0}^{t-1} \delta^{-\tau}} \sum_{\tau=0}^{t-1} \delta^{-\tau} \|\mathbf{u}_i^t - \mathbf{u}_i^\tau\|_*
$$

$$
\leq \frac{1}{\sum_{\tau=0}^{t-1} \delta^{-\tau}} \sum_{\tau=0}^{t-1} \delta^{-\tau} \sum_{q=\tau}^{t-1} \left\| \mathbf{u}_i^{q+1} - \mathbf{u}_i^q \right\|_*
$$

$$
= \frac{1}{\sum_{\tau=0}^{t-1} \delta^{-\tau}} \sum_{q=0}^{t-1} \left\| \mathbf{u}_i^{q+1} - \mathbf{u}_i^q \right\|_* \sum_{\tau=0}^{q} \delta^{-\tau}
$$

$$
= \frac{1}{\sum_{\tau=0}^{t-1} \delta^{-\tau}} \sum_{q=0}^{t-1} \left\| \mathbf{u}_i^{q+1} - \mathbf{u}_i^q \right\|_* \delta^{-q}\frac{1 - \delta^{q+1}}{1 - \delta}
$$

$$
\leq \frac{1}{1-\delta} \frac{1}{\sum_{\tau=0}^{t-1} \delta^{-\tau}} \sum_{q=0}^{t-1} \delta^{-q} \left\| \mathbf{u}_i^{q+1} - \mathbf{u}_i^q \right\|_*
$$

By Cauchy-Schwarz:

$$
\left( \frac{1}{1-\delta} \frac{1}{\sum_{\tau=0}^{t-1} \delta^{-\tau}} \sum_{q=0}^{t-1} \delta^{-q} \left\| \mathbf{u}_i^{q+1} - \mathbf{u}_i^q \right\|_* \right)^2 = \frac{1}{(1-\delta)^2} \frac{1}{\left(\sum_{\tau=0}^{t-1} \delta^{-\tau}\right)^2} \left( \sum_{q=0}^{t-1} \delta^{-q/2} \cdot \delta^{-q/2} \left\| \mathbf{u}_i^{q+1} - \mathbf{u}_i^q \right\|_* \right)^2
$$

$$
\leq \frac{1}{(1-\delta)^2} \frac{1}{\left(\sum_{\tau=0}^{t-1} \delta^{-\tau}\right)^2} \sum_{q=0}^{t-1} \delta^{-q} \cdot \sum_{q=0}^{t-1} \delta^{-q} \left\| \mathbf{u}_i^{q+1} - \mathbf{u}_i^q \right\|_*^2
$$

$$
= \frac{1}{(1-\delta)^2} \frac{1}{\sum_{\tau=0}^{t-1} \delta^{-\tau}} \sum_{q=0}^{t-1} \delta^{-q} \left\| \mathbf{u}_i^{q+1} - \mathbf{u}_i^q \right\|_*^2
$$

$$
= \frac{1}{(1-\delta)^2} \frac{1}{\sum_{\tau=0}^{t-1} \delta^{t-\tau}} \sum_{q=0}^{t-1} \delta^{t-q} \left\| \mathbf{u}_i^{q+1} - \mathbf{u}_i^q \right\|_*^2
$$

$$
\leq \frac{1}{\delta(1-\delta)^2} \sum_{q=0}^{t-1} \delta^{t-q} \left\| \mathbf{u}_i^{q+1} - \mathbf{u}_i^q \right\|_*^2
$$

Combining we get:

$$\left\| \mathbf{u}_i^t - \frac{1}{\sum_{\tau=0}^{t-1} \delta^{-\tau}} \sum_{\tau=0}^{t-1} \delta^{-\tau} \mathbf{u}_i^\tau \right\|_*^2 \leq \frac{1}{\delta(1-\delta)^2} \sum_{q=0}^{t-1} \delta^{t-q} \left\| \mathbf{u}_i^{q+1} - \mathbf{u}_i^q \right\|_*^2$$

Summing over all $t$ and re-arranging we get:

$$\sum_{t=1}^{T} \left\| \mathbf{u}_i^t - \frac{1}{\sum_{\tau=0}^{t-1} \delta^{-\tau}} \sum_{\tau=0}^{t-1} \delta^{-\tau} \mathbf{u}_i^\tau \right\|_*^2 \leq \frac{1}{\delta(1-\delta)^2} \sum_{t=1}^{T} \sum_{q=0}^{t-1} \delta^{t-q} \left\| \mathbf{u}_i^{q+1} - \mathbf{u}_i^q \right\|_*^2$$

$$= \frac{1}{\delta(1-\delta)^2} \sum_{q=0}^{T-1} \delta^{-q} \left\| \mathbf{u}_i^{q+1} - \mathbf{u}_i^q \right\|_*^2 \sum_{t=q+1}^{T} \delta^t$$

$$= \frac{1}{\delta(1-\delta)^2} \sum_{q=0}^{T-1} \delta^{-q} \left\| \mathbf{u}_i^{q+1} - \mathbf{u}_i^q \right\|_*^2 \frac{\delta(\delta^q - \delta^T)}{1-\delta}$$

$$= \frac{1}{(1-\delta)^3} \sum_{q=0}^{T-1} \left\| \mathbf{u}_i^{q+1} - \mathbf{u}_i^q \right\|_*^2 (1 - \delta^{T-q})$$

$$\leq \frac{1}{(1-\delta)^3} \sum_{q=0}^{T-1} \left\| \mathbf{u}_i^{q+1} - \mathbf{u}_i^q \right\|_*^2$$

∎

# F   Proof of Theorem 14

**Theorem 14.**   *Algorithm $\mathcal{A}'$ achieves regret at most the minimum of the following two terms:*

$$\sum_{t=1}^{T} \left\langle \mathbf{w}_i^* - \mathbf{w}_i^t, \mathbf{u}_i^t \right\rangle \leq \log(T) \left( 2 + \frac{\alpha}{\eta_*} + (2 + \eta_* \cdot \beta) \sum_{t=1}^{T} \|\mathbf{u}_i^t - \mathbf{u}_i^{t-1}\|_*^2 \right) - \frac{\gamma}{\eta_*} \sum_{t=1}^{T} \|\mathbf{w}_i^t - \mathbf{w}_i^{t-1}\|^2;$$

$$\sum_{t=1}^{T} \left\langle \mathbf{w}_i^* - \mathbf{w}_i^t, \mathbf{u}_i^t \right\rangle \leq \log(T) \left( 1 + \frac{\alpha}{\eta_*} + (1 + \alpha \cdot \beta) \cdot \sqrt{2 \sum_{t=1}^{T} \|\mathbf{u}_i^t - \mathbf{u}_i^{t-1}\|_*^2} \right)$$

*Proof.* We break the proof in the two corresponding parts.

**First part.**   Consider a round $r$ and let $T_r$ be its final iteration. Also let $I_r = \sum_{t=1}^{T_r} \|\mathbf{u}_i^t - \mathbf{u}_i^{t-1}\|_*^2$. First observe that by the definition of $B_r$:

$$\frac{1}{2} I_r \leq B_r \leq 2 \cdot I_r + 1 \tag{18}$$

By the definition of $\eta$, we know that

$$\frac{1}{\eta_*} \leq \frac{1}{\eta} < \frac{1}{\eta_*} + \frac{\sqrt{B_r}}{\alpha}. \tag{19}$$

By the regret guarantee of algorithm $\mathcal{A}(\eta_r)$, we have that:

$$\sum_{t=T_{r-1}+1}^{T_r} \left\langle \mathbf{w}_i^* - \mathbf{w}_i^t, \mathbf{u}_i^t \right\rangle \leq \frac{\alpha}{\eta} + \eta \cdot \beta \sum_{t=T_{r-1}+1}^{T_r} \|\mathbf{u}_i^t - \mathbf{u}_i^{t-1}\|_*^2 - \frac{\gamma}{\eta} \sum_{t=T_{r-1}+1}^{T_r} \|\mathbf{w}_i^t - \mathbf{w}_i^{t-1}\|^2$$

$$\leq \frac{\alpha}{\eta_*} + \sqrt{B_r} + \eta_* \cdot \beta \sum_{t=T_{r-1}+1}^{T_r} \|\mathbf{u}_i^t - \mathbf{u}_i^{t-1}\|_*^2 - \frac{\gamma}{\eta_*} \sum_{t=T_{r-1}+1}^{T_r} \|\mathbf{w}_i^t - \mathbf{w}_i^{t-1}\|^2$$

$$\leq \frac{\alpha}{\eta_*} + \sqrt{B_r} + \eta_* \cdot \beta \sum_{t=1}^{T} \|\mathbf{u}_i^t - \mathbf{u}_i^{t-1}\|_*^2 - \frac{\gamma}{\eta_*} \sum_{t=T_{r-1}+1}^{T_r} \|\mathbf{w}_i^t - \mathbf{w}_i^{t-1}\|^2$$

Since $\sqrt{B_r} \leq B_r + 1 \leq 2 \cdot I_r + 2$:

$$\sum_{t=T_{r-1}+1}^{T_r} \left\langle \mathbf{w}_i^* - \mathbf{w}_i^t, \mathbf{u}_i^t \right\rangle \leq \frac{\alpha}{\eta_*} + 2 + (2 + \eta_* \cdot \beta) \sum_{t=1}^{T} \|\mathbf{u}_i^t - \mathbf{u}_i^{t-1}\|_*^2 - \frac{\gamma}{\eta_*} \sum_{t=T_{r-1}+1}^{T_r} \|\mathbf{w}_i^t - \mathbf{w}_i^{t-1}\|^2$$

Since at each round we are doubling the bound $B_r$ and since $\sum_{t=1}^{T} \|\mathbf{u}_i^t - \mathbf{u}_i^{t-1}\|_*^2 \leq T$, there are at most $\log(T)$ rounds. Summing up the above inequality for each of the at most $\log(T)$ rounds, yields the claimed bound in Equation (4).

**Second part.** Again consider any round $r$. By Equations (18), (19), the fact that $\eta \leq \frac{\alpha}{\sqrt{B_r}} \leq \frac{\alpha\sqrt{2}}{\sqrt{I_r}}$ and by the regret of algorithm $\mathcal{A}(\eta_r)$:

$$\begin{aligned}
\sum_{t=T_{r-1}+1}^{T_r} \left\langle \mathbf{w}_i^* - \mathbf{w}_i^t, \mathbf{u}_i^t \right\rangle &\leq \frac{\alpha}{\eta_*} + \sqrt{B_r} + \eta \cdot \beta \sum_{t=T_{r-1}+1}^{T_r} \|\mathbf{u}_i^t - \mathbf{u}_i^{t-1}\|_*^2 \\
&\leq \frac{\alpha}{\eta_*} + \sqrt{B_r} + \eta \cdot \beta \cdot I_r \\
&\leq \frac{\alpha}{\eta_*} + \sqrt{B_r} + \alpha \cdot \beta \cdot \sqrt{2I_r} \\
&\leq \frac{\alpha}{\eta_*} + \sqrt{2I_r + 1} + \alpha \cdot \beta \cdot \sqrt{2I_r} \\
&\leq \frac{\alpha}{\eta_*} + 1 + \sqrt{2I_r} + \alpha \cdot \beta \cdot \sqrt{2I_r} \\
&\leq \frac{\alpha}{\eta_*} + 1 + (1 + \alpha \cdot \beta)\sqrt{2\sum_{t=1}^{T} \|\mathbf{u}_i^t - \mathbf{u}_i^{t-1}\|_*^2}
\end{aligned}$$

Again since the number of rounds is at most $\log(T)$, by summing up the above bound for each round $r$, we get the second part of the theorem. ∎

## G  Proof of Corollary 16

**Corollary 16.** *If $\mathcal{A}$ satisfies the RVU($\rho$) property, and also $\|\mathbf{w}_i^t - \mathbf{w}_i^{t-1}\| \leq \kappa\rho$, then $\mathcal{A}'$ with $\eta_* = T^{-1/4}$ achieves regret $\tilde{O}(T^{1/4})$ if played against itself, and $\tilde{O}(\sqrt{T})$ against any opponent.*

*Proof.* Observe that at any round of $\mathcal{A}'$, algorithm $\mathcal{A}$ is run with $\eta_r \leq \eta_*$. Thus by the property of algorithm $\mathcal{A}$, we have that at every iteration: $\|\mathbf{w}_i^t - \mathbf{w}_i^{t-1}\| \leq \kappa\eta_* = \kappa T^{-1/4}$. If all players use algorithm $\mathcal{A}'$, then by similar reasoning as in Theorem 4 we know that:

$$\|\mathbf{u}_i^t - \mathbf{u}_i^{t-1}\|_*^2 \leq (n-1)\sum_{j\neq i} \|\mathbf{w}_j^t - \mathbf{w}_j^{t-1}\|^2 \leq (n-1)^2 \gamma^2 \eta_*^2 = (n-1)^2 \kappa^2 T^{-1/2}$$

Hence, by Equation 5, the regret of each player is bounded by:

$$\begin{aligned}
\sum_{t=1}^{T} \left\langle \mathbf{w}_i^* - \mathbf{w}_i^t, \mathbf{u}_i^t \right\rangle &\leq \log(T)\left( \frac{\alpha}{\eta_*} + (1 + \alpha \cdot \beta) \cdot \sqrt{\sum_{t=1}^{T} \|\mathbf{u}_i^t - \mathbf{u}_i^{t-1}\|_*^2} \right) \\
&\leq \log(T)\left( \alpha T^{1/4} + (1 + \alpha \cdot \beta) \cdot \sqrt{T \cdot (n-1)^2 \kappa^2 T^{-1/2}} \right) \\
&= \log(T)\left( \alpha T^{1/4} + (1 + \alpha \cdot \beta) \cdot (n-1)\kappa T^{1/4} \right) = \tilde{O}(T^{1/4})
\end{aligned}$$

∎

# H   Fast convergence via a first order regret bound for cost-minimization

In this section, we show how a different regret bound can also lead to a fast convergence rate for a smooth game. For some technical reasons we consider cost instead of utility throughout this section. We use $c_i : S_1 \times \ldots \times S_n \to [0,1]$ to denote the cost function, and similarly to previous sections $C(\mathbf{s}) = \sum_{i \in N} c_i(\mathbf{s}), C(\mathbf{w}) = \mathbb{E}_{\mathbf{s} \sim \mathbf{w}}[C(\mathbf{s})], \mathrm{OPT}' = \min_{\mathbf{s} \in S_1 \times \ldots \times S_n} C(\mathbf{s})$. A game is $(\lambda, \mu)$-smooth if there exists a strategy profile $\mathbf{s}^*$, such that for any strategy profile $\mathbf{s}$:

$$\sum_{i \in N} c_i(s_i^*, \mathbf{s}_{-i}) \leq \lambda \mathrm{OPT}' + \mu C(\mathbf{s}). \tag{20}$$

Now suppose each player $i$ uses a no-regret algorithm to produce $\mathbf{w}_i^t$ on each round and receives cost $c_{i,s}^t = \mathbb{E}_{\mathbf{s}_{-i} \sim \mathbf{w}_{-i}^t}[c_i(s, \mathbf{s}_{-i})]$ for each strategy $s \in S_i$. Moreover, for any fixed strategy $s$, the no-regret algorithm ensures

$$\sum_{t=1}^{T} \left\langle \mathbf{w}_i^t, \mathbf{c}_i^t \right\rangle - \sum_{t=1}^{T} c_{i,s}^t \leq A_1 \sqrt{\log d \left( \sum_{t=1}^{T} c_{i,s}^t \right)} + A_2 \log d \tag{21}$$

for some absolute constants $A_1$ and $A_2$. Note that this form of first order bound can be achieved by a variety of algorithms such as Hedge with appropriate learning rate tuning. Under this setup, we prove the following:

**Theorem 23.** *If a game is $(\lambda, \mu)$-smooth and each player uses a no-regret algorithm with a regret satisfying Eq.* (21)*, then we have*

$$\frac{1}{T} \sum_{t=1}^{T} C(\mathbf{w}^t) \leq \frac{\lambda(1+\mu)}{\mu(1-\mu)} \mathrm{OPT}' + \frac{An \log d}{T}$$

*where $A = \frac{A_1^2 \mu}{(1-\mu)^2} + \frac{2A_2}{1-\mu}$.*

*Proof.* Using the regret bound and Cauchy-Schwarz inequality, we have

$$\sum_{t=1}^{T} C(\mathbf{w}^t) = \sum_{t=1}^{T} \sum_{i \in N} \left\langle \mathbf{w}_i^t, \mathbf{c}_i^t \right\rangle$$

$$\leq \sum_{t=1}^{T} \sum_{i \in N} c_{i,s_i^*}^t + A_1 \sqrt{\log d} \sum_{i \in N} \sqrt{\sum_{t=1}^{T} c_{i,s_i^*}^t} + A_2 n \log d$$

$$\leq \sum_{t=1}^{T} \sum_{i \in N} c_{i,s_i^*}^t + A_1 \sqrt{n \log d} \sqrt{\sum_{T=1}^{T} \sum_{i \in N} c_{i,s_i^*}^t} + A_2 n \log d. \tag{22}$$

By the smoothness assumption, we have

$$\sum_{i \in N} c_{i,s_i^*}^t = \mathbb{E}_{\mathbf{s} \sim \mathbf{w}^t} \left[ \sum_{i \in N} c_i(s_i^*, \mathbf{s}_{-i}) \right] \leq \lambda \mathrm{OPT}' + \mu \mathbb{E}_{\mathbf{s} \sim \mathbf{w}^t}[C(\mathbf{s})] = \lambda \mathrm{OPT}' + \mu C(\mathbf{w}^t),$$

and therefore $\sum_{t=1}^{T} \sum_{i \in N} c_{i,s_i^*}^t \leq x^2$ where we define $x = \sqrt{\lambda T \mathrm{OPT}' + \mu \sum_{t=1}^{T} C(\mathbf{w}^t)}$. Now applying this bound in Eq. (22), we continue with

$$\frac{1}{\mu} \left( x^2 - \lambda T \mathrm{OPT}' \right) \leq x^2 + (A_1 \sqrt{n \log d})x + A_2 n \log d.$$

Rearranging gives a quadratic inequality $ax^2 + bx + c \leq 0$ with

$$a = \frac{1-\mu}{\mu}, \quad b = -A_1 \sqrt{n \ln d}, \quad c = -\frac{\lambda}{\mu} T \mathrm{OPT}' - A_2 n \log d,$$

and solving for $x$ gives

$$x \leq \frac{\mu}{2(1-\mu)}(-b + \sqrt{b^2 - 4ac}) \leq \frac{\mu}{1-\mu} \sqrt{b^2 - 2ac}.$$

Finally solving for $\sum_{t=1}^{T} C(\mathbf{w}^t)$ (hidden in the definition of $x$) gives the bound stated in the theorem. ∎

Note that the price of total anarchy is larger than the one achieved by previous analysis by a multiplicative factor of $1 + \frac{1}{\mu}$, but the convergence rate is much faster ($n$ times faster compared to optimistic mirror descent or optimistic FTRL).

# I $\Omega(\sqrt{T})$ Lower Bounds on Regret for other Dynamics

We consider a two-player zero-sum game which can be described by a utility matrix $A$. Assume the row player uses MWU with a fixed learning rate $\eta$, and the column player plays the best response, that is, a pure strategy that minimizes the row player's expected utility for the current round. Then the following theorem states that no matter how $\eta$ is set, there is always a game $A$ such that the regret of the row player is at least $\Omega(\sqrt{T})$.

**Theorem 24.** *In the setting described above, let $r(T)$ and $r'(T)$ be the regret of the row player for the game $A = \begin{pmatrix} 1 & 0 \\ 0 & 1 \end{pmatrix}$ and $A' = \begin{pmatrix} 1 \\ 0 \end{pmatrix}$ respectively after $T$ rounds. Then $\max\{r(T), r'(T)\} \geq \Omega(\sqrt{T})$.*

*Proof.* For game $A$, according to the setup, one can verify that the row player will play a uniform distribution and receive utility $\frac{1}{2}$ on round $t$ where $t$ is odd, and for the next round $t+1$, the row player will put slightly more weights on one row and the column player will pick the column that has 0 utility for that row. Specifically, the expected utility of the row player is $\frac{e^{\eta(t-1)/2}}{e^{\eta(t-1)/2}+e^{\eta(t+1)/2}} = \frac{1}{1+e^\eta}$. Therefore, the regret is (assuming $T$ is even for simplicity)

$$r(T) = \frac{T}{2} - \frac{T}{2}\left(\frac{1}{2} + \frac{1}{1+e^\eta}\right) = \frac{T}{2} \cdot \frac{e^\eta - 1}{e^\eta + 1}.$$

For game $A'$, the expected utility of the row player on round $t$ is $\frac{e^{\eta(t-1)}}{e^{\eta(t-1)}+1}$, and thus the regret is

$$r'(T) = T - \sum_{t=1}^{T} \frac{e^{\eta(t-1)}}{e^{\eta(t-1)} + 1} = \sum_{t=1}^{T} \frac{1}{e^{\eta(t-1)} + 1} \geq \sum_{t=1}^{T} \frac{1}{2e^{\eta(t-1)}} = \frac{1 - e^{-T\eta}}{2(1 - e^{-\eta})}.$$

Now if $\eta \geq 1$, then $r(T) \geq \frac{T}{2} \cdot \frac{e-1}{e+1} = \Omega(T)$. If $\eta \leq \frac{1}{T}$, then $r'(T) \geq \frac{1-e^{-1}}{2(1-e^{-\frac{1}{T}})} \geq \frac{T(1-e^{-1})}{2} = \Omega(T)$. Finally when $\frac{1}{T} \leq \eta \leq 1$, we have

$$r(T) + r'(T) \geq \frac{T}{2} \cdot \frac{e^\eta - 1}{e+1} + \frac{1 - e^{-1}}{2(1 - e^{-\eta})} \geq \frac{T}{2} \cdot \frac{e^\eta - 1}{e+1} + \frac{1 - e^{-1}}{2(e^\eta - 1)} \geq \sqrt{T \cdot \frac{1 - e^{-1}}{e+1}} = \Omega(\sqrt{T}).$$

To sum up, we have $\max\{r(T), r'(T)\} \geq \Omega(\sqrt{T})$. ∎