[Reviews · NeurIPS 2015]

Submitted by Assigned_Reviewer_1

The authors perform theoretical analysis about faster convergence with multi-player normal-form games by generalizing techniques for two-player zero-sum games. They also perform empirical evaluation by using the 4-bidder simultaneous auction game.

The paper is understandable and contains nice theoretical and empirical results. My only comment is that it would be better to use various n and m for performance evaluation (i.e. not limited to 4-bidders) to more clearly elucidate the behavior of Hedge and Optimistic Hedge.
Summary: The paper has solid theoretical results as well as nice experimental results in the 4-bidder simultaneous auction game.

Submitted by Assigned_Reviewer_2

The paper is concerned with two problems:

(1)

How does the social welfare of players using regret minimization algorithms compare to the optimal welfare. (2)

Can one obtain better regret bounds when all players use a regret minimization algorithm

It has been shown in the past that the social welfare converges at a rate of 1/T to the optimal welfare for zero-sum games. However, the question remained open for general non-zero sum games. The authors show that under some smoothness assumptions on the game (introduced by Roughgarden) the same rate of convergence can be achieved when all players use a regret minimization algorithm that has the so called RVU property. Moreover, the authors show that the optimistic mirror descent algorithm of Rahklin admits this property. The authors also show that an optimistic variation of FTRL (OFTRL) with a recency bias also admits this property.

For problem (2), the authors show that the regret of each individual player is in O(T^{1/4}) when using their variation of FTRL and this can be seen also in their simulations. Furthermore, they enhance their algorithm by using a doubling trick in such a way that if all players use OFTRL, then their regret grows like T^{1/4} and yet the regret does not grow faster than O(T^1/2) if the players observe adversarial rewards.

The paper is really well written and solves an interesting problem. I particularly liked the generality of the result in terms of the RVU property.
Summary: The paper deals with bounds on regret minimization algorithms in games. The usual regret bounds on these algorithms is in O(\sqrt(T)). However, this assumes that the learner faces a completely adversarial opponent. However, it is natural to assume that on a game everyone will play a regret minimization algorithm and the question is whether or not one can obtain better rates in this scenario. The authors show that regret in O(T^{1/4}) is achievable for general games.

Submitted by Assigned_Reviewer_3

The paper introduces a new method for regret minimization in multiplayer normal-form games. Specifically, the paper generalizes previous algorithms (Optimistic Mirror Descent and Optimistic Follow the Regularized Leader) for two-player zero-sum games to multiplayer general-sum games.

The experimental results compared to Hedge is impressive. The content of the paper would appeal to the NIPS community.

I have some specific questions and comments:

- pg. 3: why is the root removed from the log when claiming "converges to O(n log(d) sqrt(T)", where in r_i(T) the root contains the log? Is this a typo?

- pg. 6: "This is the first fast convergence result to CCE using natural, decoupled no-regret dynamics." I have two problems with this statement: 1. "natural" and "decoupled" are not clearly defined; and how is it "decoupled" anyway-- Corollary 12 upon which this claim is base requires all players to use OFTRL with specific choices of M_i^t and \eta. 2. It is a bit strong: what about Hart & Mas-Colell's regret-matching (i.e. [13])? Either clarify, rephrase, or remove this claim.

Some things to think about:

- Any ideas or comments on whether these results can be extended in a straight-forward way to the partial information setting using sampling (e.g. Optimistic Exp3)?

Some minor points for camera-ready if accepted: - pg. 1: ".. a chink that hints", what is "chink"? - pg. 3: Remove the space after PoA in "(PoA )" - pg. 4: end of Thm4 proof, is there a max_{j \in N} missing in the right-most inequality? Maybe just remove the j subscript after removing the \sum_{j \neq i}? - Fix argmax so that the set being maxed over is entirely under the argmax (i.e. arg is not separated from max) - A number of items in the bibliography need to be fixed. Volume, number, pages are missing for [4], write out the full conference names for [19] and [20].
Summary: This is a well-written paper with some significant new results in the difficult setting of no-regret learning in multiplayer general-sum games. The paper generalizes previous results, shows that when players use the same algorithm (with the RVU property) that they can enjoy faster convergence than in the adversarial case, but also show that in the adversarial case the worst-case O(log(T) sqrt(T)) is still attainable when using a parameterized step-size and the doubling trick.

Author Feedback
Author rebuttal: We thank the reviewers for their thoughtful comments and careful reading of the paper and we address some of their questions below.

With respect to reviewer 3 questions:
1) Indeed the log(d) should be under the square root in the first rate of convergence, that is a typo.

2) A formal model of decoupled and decentralized computation was defined in Daskalakis, Deckelbaum, Kim "Near-optimal no-regret algorithms for zero-sum games". This is the computational model we had in mind. Regret matching is an example of such a model but it converges at a slower rate of sqrt{T}. We will certainly make the comment more precise by referring to this computational model.

3) Other minor typos that you point out will also be addressed in the revision, thanks for noticing these!

We also want to point out that the extension of our main result in Section 3.2 (see end of the section), which states that one can also get convergence to correlated rather than coarse correlated equilibria by invoking the black-box reduction of Blum and Mansour does not hold. The black-box reduction of Blum and Mansour cannot be invoked as is for our results to extend to correlated equilibria as it does not preserve some required stability properties. We note that this does not affect any of the main results in the paper, and we will remove that remark from the final version.

Once again we thank all the reviewers for their feedback and will address all the comments in the final version of the paper.